# Pauses in cholinergic interneuron firing exert an inhibitory control on striatal output in vivo

Stefano Zucca, Aya Zucca, Takashi Nakano, Sho Aoki, Jeffery Wickens*

Okinawa Institute of Science and Technology Graduate University, Okinawa, Japan

**Abstract** The cholinergic interneurons (CINs) of the striatum are crucial for normal motor and behavioral functions of the basal ganglia. Striatal CINs exhibit tonic firing punctuated by distinct pauses. Pauses occur in response to motivationally significant events, but their function is unknown. Here we investigated the effects of pauses in CIN firing on spiny projection neurons (SPNs) – the output neurons of the striatum – using in vivo whole cell and juxtacellular recordings in mice. We found that optogenetically-induced pauses in CIN firing inhibited subthreshold membrane potential activity and decreased firing of SPNs. During pauses, SPN membrane potential fluctuations became more hyperpolarized and UP state durations became shorter. In addition, short-term plasticity of corticostriatal inputs was decreased during pauses. Our results indicate that, in vivo, the net effect of the pause in CIN firing on SPNs activity is inhibition and provide a novel mechanism for cholinergic control of striatal output.

DOI: https://doi.org/10.7554/eLife.32510.001

## Introduction

Acetylcholine – the first neurotransmitter to be identified – is present in high concentrations in the striatum of the basal ganglia (*Zhou et al., 2001*) where it is released by a population of intrinsic cholinergic interneurons (CINs) that are distinct from the cholinergic projection neurons that innervate the cortex (*Woolf and Butcher, 1981*). In addition, recent studies indicate an external cholinergic innervation from the pedunculopontine and laterodorsal tegmental nuclei to the dorsal striatum and nucleus accumbens (*Dautan et al., 2016*, *2014*). CINs play an important role in behavior by modulating the activity of the principal output neurons of the striatum, the spiny projection neurons (SPNs) (*Witten et al., 2010*). Previously there has been great interest in the role of CINs in disorders of the basal ganglia (*Maurice et al., 2015*; *Pisani et al., 2007*; *Shen et al., 2015*) and more recently in behavioral flexibility (*Aoki et al., 2015*; *Bradfield et al., 2013*; *Okada et al., 2014*). The neuronal mechanisms for regulation of SPN activity by CINs are therefore of intense current interest.

CINs are tonically active (*Wilson et al., 1990*) and although they represent only 1% of striatal neurons their extensive axonal fields allow them to control large striatal regions (*Bolam et al., 1984*). These features combine with a diverse array of receptors for acetylcholine both at pre- and post-synaptic locations (*Hernández-Echeagaray et al., 1998*; *Hersch et al., 1994*; *Sugita et al., 1991*). Thus, CINs are crucial modulators of striatal function. In particular, CINs control SPN excitability and response to glutamatergic inputs via activation of postsynaptic muscarinic type 1 receptors (M1). At the presynaptic level CINs exert a tonic inhibitory influence over incoming information via the activation of muscarinic type 2 (M2/M4) receptors located on cortical and thalamic glutamatergic terminals.

The firing pattern of CINs in vivo varies according to behavioural context. During slow wave activity or activated brain states, spontaneous CIN activity ranges from tonic firing, which may be regular or irregular, to bursts of firing separated by pauses of variable duration (*Sharott et al., 2012*). In

*For correspondence:
wickens@oist.jp

Competing interests: The authors declare that no competing interests exist.

**eLife digest** Nerve cells or neurons communicate with one another using electrical impulses and chemical messengers called neurotransmitters. Additional molecules known as neuromodulators regulate the communication process. In contrast to neurotransmitters, neuromodulators do not send messages directly from one neuron to the next. Instead they change the way that neurons respond to neurotransmitters.

For example, the neuromodulator acetylcholine is most abundant in a region called the striatum. Located deep within the brain, the striatum contributes to learning and memory, motivation, and movement. Studies in rodents show that neurons within the striatum called cholinergic interneurons are almost continuously active. Each time these cells fire, they release acetylcholine. But whenever an animal experiences something unusual or important, the interneurons temporarily stop firing. Zucca et al. wanted to know whether these pauses in firing also act as a signal within the striatum.

To find out, Zucca et al. inserted a light-sensitive ion channel into cholinergic interneurons in the mouse striatum. Activating the ion channels with a laser beam stopped the interneurons from firing. Zucca et al. showed that these pauses in firing reduced the activity of another group of neurons, the spiny projection neurons. These are the major output neurons of the striatum. They send messages from the striatum to other parts of the brain. The results thus suggest that cholinergic interneurons signal notable events by temporarily blocking output from the striatum.

Understanding how cholinergic interneurons work will help reveal how the striatum drives behavior. It may also lead to treatments for diseases caused by cholinergic system dysfunction. Many patients with Parkinson's disease or schizophrenia take medicines to block the effects of acetylcholine. Understanding how acetylcholine affects the striatum may help clarify how these treatments work.

DOI: https://doi.org/10.7554/eLife.32510.002

awake animals exposed to repeated, motivationally significant stimuli, putative CINs respond by a pause in firing (*Graybiel et al., 1994*; *Kimura et al., 1984*; *Morris et al., 2004*), sometimes preceded and often followed by a burst. Previous work (*Crittenden et al., 2017*; *Ding et al., 2010*; *Doig et al., 2014*; *English et al., 2011*; *Nelson et al., 2014*) has focused on the effects of the burst component. In awake animals, the pause has been reported to have either no reliable effects with pauses of short duration (*English et al., 2011*), or a mixture of excitation and inhibition during pauses of long duration (*Witten et al., 2010*) on firing activity of SPNs recorded extracellularly. Here we focus on the effects of the pause on subthreshold membrane potential fluctuations of SPNs recorded intracellularly, as well as suprathreshold firing activity of SPNs identified by juxtacellular recording in vivo.

We used optogenetic silencing of CINs – optimized to produce a pause in nearby CINs – during whole-cell and juxtacellular electrophysiological recording from CINs and SPNs in vivo. We found that a pause in CIN firing caused an inhibitory effect on SPN activity in vivo, reversing expectations based on in vitro studies and computational theories (*Ashby and Crossley, 2011*; *Franklin and Frank, 2015*; *Pakhotin and Bracci, 2007*) and complementing previous in vivo findings (*English et al., 2011*; *Witten et al., 2010*). In particular, a pause in CIN firing hyperpolarizes the membrane potential of SPNs and reduces the short-term plasticity at cortical inputs, leading to decreased firing of SPNs in vivo.

## Results

### Pause in cholinergic interneuron firing decreases firing of striatal projection neurons in vivo.

We injected an adeno-associated virus (AAV) encoding the neuronal silencer Halorhodopsin (NpHR) into the striatum of ChAT-cre mice to induce pauses in CINs that mimic naturally occurring pauses (*Figure 1A*). Histological analysis confirmed that the expression of NpHR was restricted to CINs in the dorsal striatum (*Figure 1B* and *Figure 1C*). To assess the specificity of viral transduction we counted cells that express NpHR-eYFP and stained for anti-ChAT. We found that the 99.0 ± 0.48%

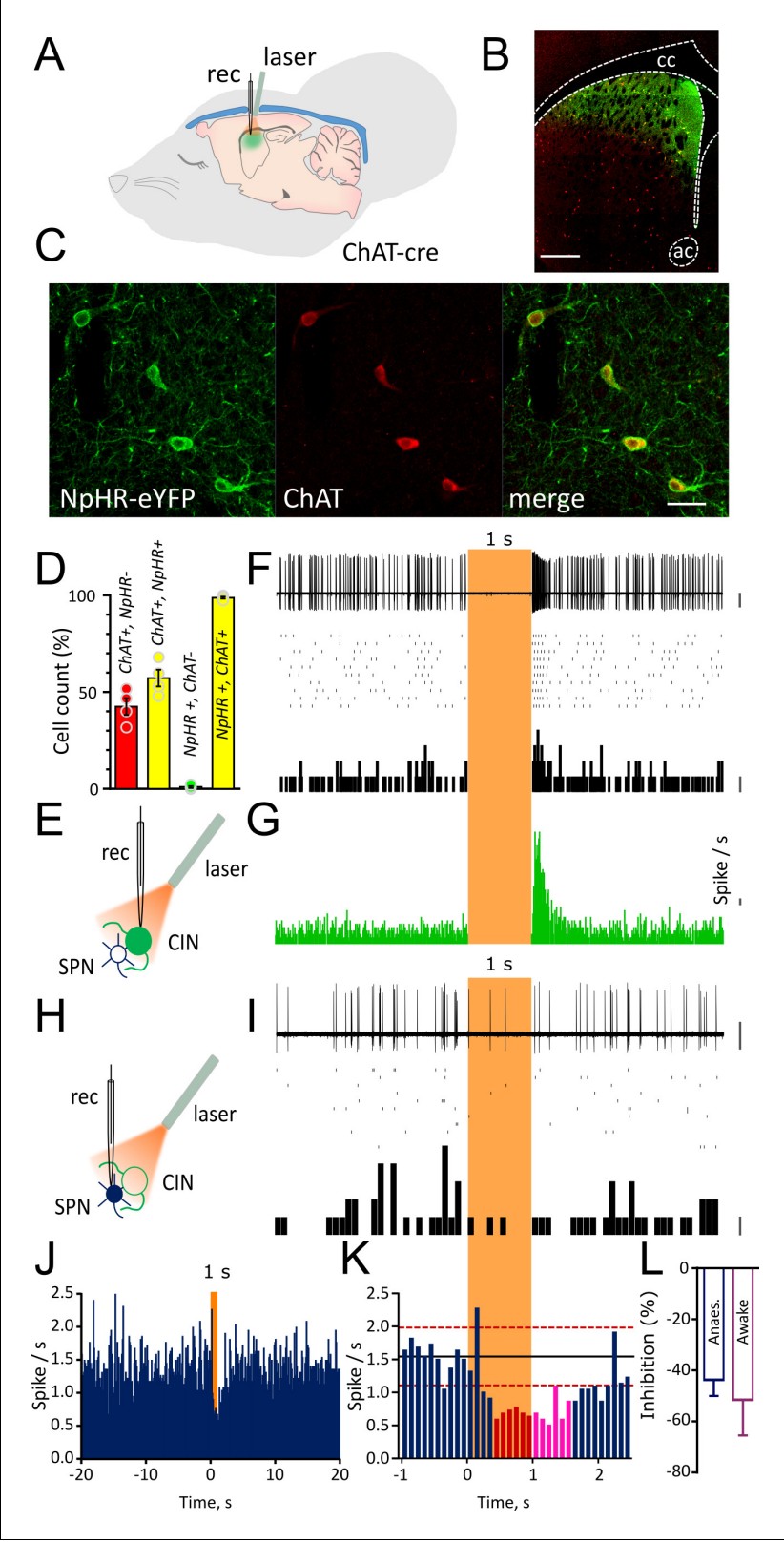

**Figure 1.** Light-induced pause in CINs inhibit spontaneous firing of SPNs in vivo. (**A**) Schematic illustration of the optogenetic approach and in vivo recordings in the anaesthetized mouse. (**B**) Confocal fluorescent image of a coronal section of the striatum confirming cell-specific expression of NpHR-eYFP (green) in ChAT positive cells (red) restricted to dorsal striatum. Scale bar = 500 µm. cc: corpus callosum; ac: anterior commissure (**C**) Magnified

*Figure 1 continued on next page*

*Figure 1 continued*

area from (**B**) showing co-localization of NpHR-eYFP (green) and anti-ChAT (red) scale bar = 50 μm. (**D**) Bar graph summarizing the quantifications to show transduction level (left two bars) and high-specificity (right two bars) of NpHR in CINs in the dorsal striatum. (**E**) Experimental configuration for recording opto-tagged CINs in vivo. (**F**) Superimposed traces of a juxtacellular recording showing light-induced pause in their spontaneous firing (top) and relative raster plot (middle) and PSTH (bottom) PSTH calibration: 1 spike/bin. (**G**) PSTH of all CINs showing complete silencing during light (1 s) and rebound firing immediately after the light. (bin = 10 ms, N/n = 14/16) PSTH calibration: 1 spike/s. (**H**) Experimental configuration for juxtacellular recordings from SPNs during light-induced pauses of CINs. (**I**) Superimposed traces of a representative unit obtained from a SPN (top) and relative raster plot (middle) and PSTH (bottom). (**J**) PSTH of all SPNs showing the inhibition of firing during 1 s light pulse (bin = 100 ms; N/n = 18/22). (**K**) Magnified PSTH from J to show onset of significant inhibition during light pulse. Black line indicates the mean number of spikes in the second before the light. Red dash line indicates ±2 s.D. of the mean number of spikes. Red and magenta indicate significant bins during light pulse and after light offset, respectively. (**L**) Average firing inhibition measured during light pulse (1 s) in anesthetized (N/n = 18/22) or awake mice (N/n = 6/7). Data are represented as mean ± SEM. N = mice; n = cells. Calibration bar for juxtacellular traces: 2 mV. Orange boxes indicate light on period.

DOI: https://doi.org/10.7554/eLife.32510.003

The following source data and figure supplements are available for figure 1:

**Source data 1.** This spreadsheet contains the quantication of the viral transduction and the firing responses to optogenetic manipulation of the neurons shown in *Figure 1*.

DOI: https://doi.org/10.7554/eLife.32510.007

**Figure supplement 1.** Optogenetic inhibition of CIN firing.

DOI: https://doi.org/10.7554/eLife.32510.004

**Figure supplement 1—source data 1.** This spreadsheet contains the firing responses of individual cholinergic neurons in slices and in vivo used to validate the optogenetic approach as shown in the graphs in *Figure 1—figure supplement 1*.

DOI: https://doi.org/10.7554/eLife.32510.008

**Figure supplement 2.** Firing properties of opto-tagged CINs and putative SPNs with juxtacellular recordings.

DOI: https://doi.org/10.7554/eLife.32510.005

**Figure supplement 2—source data 1.** This spreadsheet contains the firing properties for the individual neurons shown in *Figure 1—figure supplement 2*.

DOI: https://doi.org/10.7554/eLife.32510.009

**Figure supplement 3.** Short pause in CINs does not inhibit spontaneous firing of SPNs in vivo.

DOI: https://doi.org/10.7554/eLife.32510.006

**Figure supplement 3—source data 1.** This spreadsheet contains the firing responses to optogenetic manipulations of the individual neurons shown in *Figure 1—figure supplement 3*.

DOI: https://doi.org/10.7554/eLife.32510.010

of the cells that expressed NpHR-eYFP also stained for anti-ChAT leaving only 1.0 ± 0.48% that were non-specifically expressing NpHR-eYFP. Our injections targeting the dorsal striatum infected 57.35 ± 4.33% of ChAT positive neurons (*Figure 1D*). Whole cell recordings obtained both in acute slices and in vivo from anaesthetized mice showed that a brief pulse of light produced a hyperpolarization of the membrane potential of CINs, causing a pause of their tonic firing (*Figure 1—figure supplement 1*). The average peristimulus time histogram (PSTH) shows that in all CINs tested, NpHR-induced inhibition caused complete silencing of firing during light (*Figure 1—figure supplement 1A and B*).

To measure the effect of the pause in CIN firing on SPN activity in vivo, we first obtained juxtacellular recordings from adjacent SPNs. Optically identified CINs were separated from putative SPNs based on their distinct firing properties and spike waveforms (*Figure 1—figure supplement 2* and *Figure 1—figure supplements 2—source data 1*). To differentiate optogenetically induced pauses from those occurring during the resetting of autonomous activity (interspike interval ranging from 138 to 902 ms) (*Figure 1—figure supplement 2* and *Figure 1—figure supplements 2—source data 1.*), we delivered a light pulse of 0.5 or 1 s which induced a complete pause of ongoing firing in all CINs (*Figure 1—figure supplement 3A* and *Figure 1G*). The pause was followed by rebound firing with a variable onset (0.5 s: 105.6 ± 11.2 ms, n = 16; 1 s: 69.37 ± 8.8 ms, n = 16) and stayed

two standard deviations (S.D.) above the mean for 79.3 ± 12.6 ms (0.5 s pause) and 138.12 ± 14.6 ms (1 s pause) (*Figure 1—figure supplement 3A* and *Figure 1G*).

The light-induced pause in CIN firing caused inhibition of firing activity in SPNs (*Figure 1I* and *Figure 1J*) which dropped below 2 s.D. from the mean firing frequency (*Figure 1K*). To calculate the significance of changes in firing rate in individual neurons, taking account of successive bins, a change in firing rate was considered significant if the cumulative sum of the observed deviation of the firing rate from the mean was less than a critical value predicted from a Poisson distribution (*Ellaway, 1978*; *Imamura and Onoda, 1983*), (see Materials and methods). In response to a 1 s exposure to light, the group average (n = 22 cells) PSTH showed a significant decrease in firing rate commencing at the 400–500 ms bin (red bins, *Figure 1K*) at the p<0.01 level (see Materials and methods). When individual cells were analyzed, within the group of cells exposed to 1 s light, a total of 14/22 cells showed a significant (p<0.05) decrease in firing, at a median latency of 700 ms from the light pulse. Three cells showed a significant (p<0.05) increase in firing after the light pulse, one cell showed a significant increase (p<0.05) followed by a significant decrease (p<0.05), and four cells showed no significant change in firing rate. However, these changes occurred at much longer latency after the light offset. To determine which responses were due to the pause (and not the rebound firing activity), we tested cells for inhibition during the light on period. A total of 9/14 of the cells showing an inhibitory response (64%) showed a significant (p<0.05) decrease within the 1 s light on period, median latency of 400 msec. Light-induced pauses of CINs resulted in a similar inhibition of SPNs firing both in anesthetized mice (1 s: 43.7 ± 6.32 % n=22) and in awake, head restrained mice (51.52 ± 13.9 % n=7) (*Figure 1L*). These results indicate that a pause in CIN firing of 1 s duration significantly reduces the firing of SPNs in vivo.

We next tested the effect of a shorter light pulse (0.5 s) on the SPNs firing. In the cells exposed to a 0.5 s light on period (n = 17 cells), the group average PSTH showed a significant decrease in firing rate at the 400–500 ms bin (red bins, *Figure 1—figure supplement 3B*). When individual cells were analyzed, however, none of the cells (0/17) showed a significant decrease during the 0.5 s period of light on. These results indicate that a pause in CIN firing of 0.5 s duration was not of sufficient duration for the effect that was observed reliably with 1 s duration.

To determine the effect of the transition from light on to light offset, we analyzed the activity of SPNs in the period after the light offset, relative to the last 500 ms of the light on period. There was no significant increase from this new baseline until 700–800 ms after the light off (p<0.01, see Materials and methods). Consistent with this, the firing rate remained more than 2 s.D. below the initial baseline mean (magenta bins, *Figure 1K*).

## Pause in cholinergic interneuron firing modulates subthreshold membrane potential fluctuations in SPNs in vivo.

To understand the cellular mechanisms underlying the reduction of firing in SPNs, we then obtained in vivo whole cell recordings of SPNs in the dorsal striatum of mice anesthetized with a ketamine/xylazine mix (*Figure 2A* and *Figure 2B*). Under these conditions SPNs showed a characteristic bimodal distribution of the membrane potential, fluctuating between a DOWN state, characterized by hyperpolarized potentials, and an UP state, where the cell is close to threshold for firing (*Figure 2B*). The best-fit of the sum of two Gaussian distributions to the all-amplitudes distribution of the membrane potential revealed that, during a pause (5 s) in CIN firing, the membrane potential of SPNs shifts to more hyperpolarized values in the DOWN state (before: −71.59 ± 1.82 mV; pause: −72.77 ± 1.86 mV; after: −71.66 ± 1.75 mV; n = 13; Repeated Measure One-way ANOVA, $F_{(2, 24)}$ =7.692; p=0.0026; *Figure 2C* and *Figure 2D*). The UP state potential was also significantly more hyperpolarized during a pause (before: −56.35 ± 1.85 mV; pause: −57.79 ± 2.01 mV; after: −55.89 ± 1.83 mV; n = 13; Repeated Measure One-way ANOVA, $F_{(2, 24)}$=6.139; p = 0.0070 *Figure 2C* and *Figure 2D*). We also observed a decrease in the number of points during the UP state, thus suggesting a shortened duration (*Figure 2C*). We then used a crossover of moving averages to detect and quantify UP and DOWN state durations. We observed significantly shorter UP states during a pause (before: 0.42 ± 0.02 s; pause: 0.39 ± 0.01 s; after: 0.41 ± 0.02 s; n = 13; Repeated Measure One-way ANOVA, $F_{(2, 24)}$=11.56, p=0.0003; *Figure 2E*) while DOWN state durations were significantly increased (before: 0.51 ± 0.03 s; pause: 0.55 ± 0.04 s; after: 0.5 ± 0.04 s; n = 13; Repeated Measure One-way ANOVA, $F_{(2,24)}$ = 5.874, p=0.0084; *Figure 2E*). The number of action potentials fired in the UP states during a pause was significantly decreased (before:

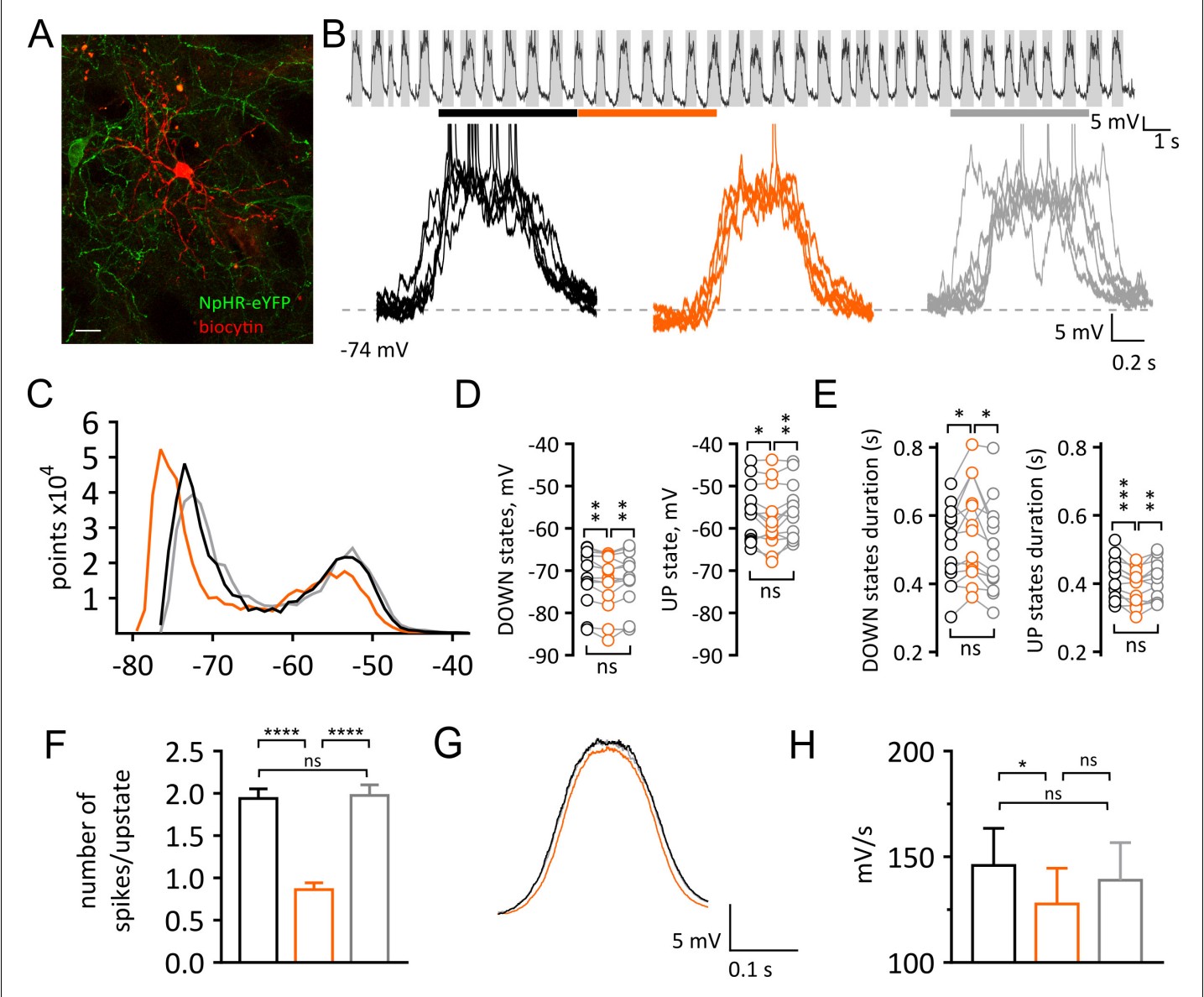

**Figure 2.** Light-induced pause in CINs alters UP and DOWN states of SPNs in vivo. (**A**) Confocal fluorescent image of a biocytin filled SPN (red) during whole cell recording and surrounding CINs expressing NpHR-eYFP (green), scale bar = 20 μm. (**B**) Top: example trace of a current clamp recording from a SPN showing UP and DOWN membrane fluctuations. Gray boxes define UP states. Bottom: superimposed UP states, each obtained from 5 s time window before, during and after a pause highlighted by black, orange, and gray lines respectively. (**C**) Representative all points histogram from SPN in (**B**) showing the effect of a pause on the distributions of membrane potentials. (**D**) Membrane potential (mV) changes during UP and DOWN states. Tukey's multiple comparisons test for Down states: before vs pause: p=0.0051; pause vs after: p=0.0084. Tukey's multiple comparisons test for UP states: before vs pause: p=0.0456; pause vs after: p=0.0071. (**E**) Changes in UP and DOWN states duration. Tukey's multiple comparisons test for DOWN states: before vs pause: p=0.0258; pause vs after: p=0.0127. Tukey's multiple comparisons test for UP states: before vs pause: p=0.0003; pause vs after: p=0.0044. (**F**) The number of action potential occurring during UP states is significantly reduced during a pause; before vs pause: p<0.0001; pause vs after: p<0.0001, by Tukey's multiple comparisons test. (**G**) Average trace from all UP states occurring before, during and after a pause, showing a slower transition to UP states during a pause. (**H**) Slope of rising phase of the UP state is significantly slower during a pause; before vs pause: p=0.0113, by Tukey's multiple comparisons test. Data are represented as mean ± SEM. (N/n = 13/13, except for (**F**) n = number of UP states, see Materials and methods.) N = mice; n = cells. APs are truncated in (**B**). Time points before, during and after a pause are indicated by black, orange and gray colors respectively; ns = not significant.

DOI: https://doi.org/10.7554/eLife.32510.011

The following source data is available for figure 2:

**Source data 1.** This spreadsheet contains the voltage responses to optogenetic manipulation for the individual neurons shown in *Figure 2*.
DOI: https://doi.org/10.7554/eLife.32510.012

1.94 ± 0.11 Hz; pause: 0.86 ± 0.08 Hz; after: 1.97 ± 0.13 Hz; n = 334, 319, 325 UP states respectively; One-way ANOVA, $F_{(2, 975)}=33.16$, p<0.0001; *Figure 2F*) UP states did not exhibit significant changes in the mean amplitude (before: 17.23 ± 1.02 mV; pause: 16.34 ± 1.27 mV; after: 16.83 ± 1.12 mV; n = 13; Repeated Measure One-way ANOVA, $F_{(2, 24)}=1.88$ p=0.1748; data not shown), nor in their frequency (before: 1.03 ± 0.04 Hz; pause: 0.98 ± 0.05 Hz; after: 1.0 ± 0.05 Hz; n = 13; Repeated Measure One-way ANOVA, $F_{(2, 24)}=2.039$, p=0.1521; data not shown). However, the transition from DOWN to UP state was significantly slower during a pause (before: 145.9 ± 17.52 mV/s; pause: 127.7 ± 16.87 mV/s; after: 139 ± 17.76 mV/s; n = 13; Repeated Measure One-way ANOVA, $F_{(2, 24)}=5.09$, p=0.0144; *Figure 2G* and *Figure 2H*). Altogether, these data indicate that a pause in CIN firing decreases the excitability of SPNs in vivo.

## Pause in cholinergic interneuron firing gates corticostriatal synaptic transmission in vivo.

The altered DOWN and UP state dynamics we observed could not be due a change in the activity of extrinsic afferents because the optogenetic stimulation and expression of NpHR is virtually specific to the striatal CINs. However, cholinergic effects on presynaptic or postsynaptic receptors are possibilities. Because UP states in the anesthetized animal represent the synchronous activation of cortical inputs we investigated if a pause in CIN firing affected the responsiveness of SPNs to repetitive synaptic inputs. We measured excitatory post-synaptic potentials (EPSPs) in response to stimulation of

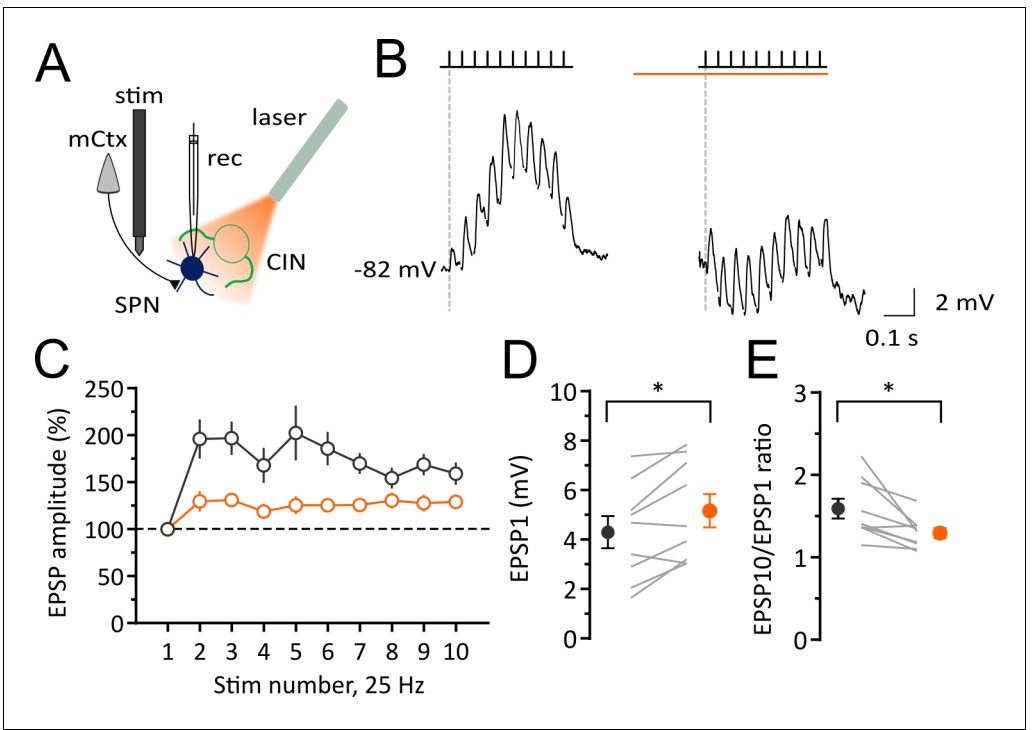

**Figure 3.** Light-induced pause in CINs decreases corticostriatal plasticity of SPNs in vivo. Experimental configuration for whole-cell recordings of evoked EPSPs in SPNs during light-induced pauses of CINs. (B) Average trace of whole-cell current-clamp recording of EPSPs elicited by cortical stimulation (10 stimuli, 25 Hz) before (left) and during a pause (right). (C) Pooled data showing that facilitation is attenuated during a pause. (D) The first EPSP amplitude is significantly increased during a pause. (E) The ratio of last and first EPSPs is significantly decreased during a pause. Lines indicate individual cells and filled circles with error bars represent mean ± SEM. (N/n = 9/9) *p<0.05 with paired *t*-test. N = mice; n = cells.
DOI: https://doi.org/10.7554/eLife.32510.013

The following source data is available for figure 3:

**Source data 1.** This spreadsheet contains the synaptic responses used to generate the graphs shown in *Figure 3*.
DOI: https://doi.org/10.7554/eLife.32510.014

glutamatergic inputs from the motor cortex in vivo (*Figure 3A*). In control conditions (before a pause in CIN firing), a train of 10 stimuli at 25 Hz, induced a gradual facilitation and summation of EPSPs over the course of the train (*Figure 3B* and *Figure 3C*). During the pause in CIN firing, we also observed a significant increase in the amplitude of the first EPSP in the stimulus train (before: $4.3 \pm 0.65$ mV; pause: $5.16 \pm 0.67$ mV; n = 9; paired *t*-test, p=0.0115; *Figure 3B* and *Figure 3D*). However, during a pause (1 s), the ratio of the last to the first EPSP, was significantly decreased (before: $1.59 \pm 0.12$; pause: $1.29 \pm 0.06$; n = 9; paired *t*-test, p=0.0146; *Figure 3B* and *Figure 3E*). These data suggest that a pause in CIN firing gates cortical information transfer at striatal synapses in vivo.

## Pause in cholinergic interneuron firing decreases intrinsic excitability of SPNs

Our finding that a pause in CIN firing modulates short term plasticity at excitatory synaptic inputs and decreases firing of SPNs in vivo could be attributed to a decrease in the intrinsic excitability of SPNs. We measured the effect of a pause on the intrinsic membrane responses of SPNs in acute slices (*Figure 4*). In agreement with previous observations (*Maurice et al., 2015*) the light-induced pause produced a significant increase in rheobase current compared to control or after light (control: $310 \pm 22.09$; light: $333.3 \pm 20.94$; after: $313.3 \pm 21.08$; n = 12; Repeated Measure One-way ANOVA, $F_{(2, 22)}=33.79$, p<0.0001; *Figure 4B*), and a significant decrease in the number of action potentials fired in response to a current pulse (n = 12; two-way ANOVA, current x light interaction, $F_{(16, 176)}=1.77$, p=0.039; *Figure 4C*). Thus, we confirmed that a pause in CIN firing decreased intrinsic excitability of SPNs.

Interruption of tonic M1 receptors activation is a possible mechanism for the decreased excitability observed during a pause. Under physiological conditions, cholinergic activation of M1 receptors increases SPN excitability, their responsiveness to cortical and thalamic inputs, and their firing by modulating multiple intracellular conductances (*Akins et al., 1990*; *Gabel and Nisenbaum, 1999*; *Galarraga et al., 1999*; *Hsu et al., 1996*; *Shen et al., 2005*, *2007*). We therefore tested the hypothesis that a pause could decrease intrinsic excitability of SPNs via modulation of M1 receptors in acute slices. We found that, similar to the effect of a pause, bath application of low concentrations of M1 receptor antagonist pirenzepine (100 nM), increased the rheobase current (control: $280 \pm 32.66$; light: $308.6 \pm 32.32$; pirenzepine: $302.9 \pm 30.99$; pirenzepine and light: $305.7 \pm 34.01$; n = 7; Repeated Measure One-way ANOVA, $F_{(3, 18)}=16.37$, p<0.0001; *Figure 4E*) and significantly shifted the relationship between current injection and number of action potential to the right (n = 7; two-way ANOVA, current x light interaction, $F_{(24, 144)}=2.519$, p=0.0004; *Figure 4F*). Moreover, the M1 receptor antagonist occluded the effect of the light, in that the combined effect of pirenzepine and light on the intrinsic excitability of SPNs was not significantly different from the individual effects (*Figure 4E*). These results suggest that interruption of M1 receptor activation during a pause is sufficient to decrease SPN excitability (*Figure 4G*).

## Discussion

Based on in vivo whole cell recordings we show that optogenetically-induced pauses in CIN firing have direct and reproducible inhibitory effects on SPN activity. Pauses in CIN firing caused hyperpolarization of subthreshold membrane potential fluctuations measured by in vivo whole-cell recording and caused a significant decrease in the firing rate of SPNs measured using in vivo juxtacellular recordings. Short-term facilitation of corticostriatal synaptic transmission was also modulated during a pause. The main novel feature of the present study is a focus on the effect of the pause uncontaminated by preceding or following bursts, which required measurement during pauses of sufficient duration to separate pause effects from effects of rebound firing. The use of in vivo whole-cell recording configuration furthermore, enabled the effects of the pause on the subthreshold membrane potential to be measured intracellularly, which revealed subthreshold changes associated with the significantly decreased firing of SPNs during the pause. In addition, in vivo whole cell recordings permitted measurement of EPSPs in response to stimulation of the motor cortex, which showed that short term plasticity of cortical input was modulated by the pause. We believe these findings constitute the first direct evidence that the pause in CIN firing carries a meaningful signal to postsynaptic

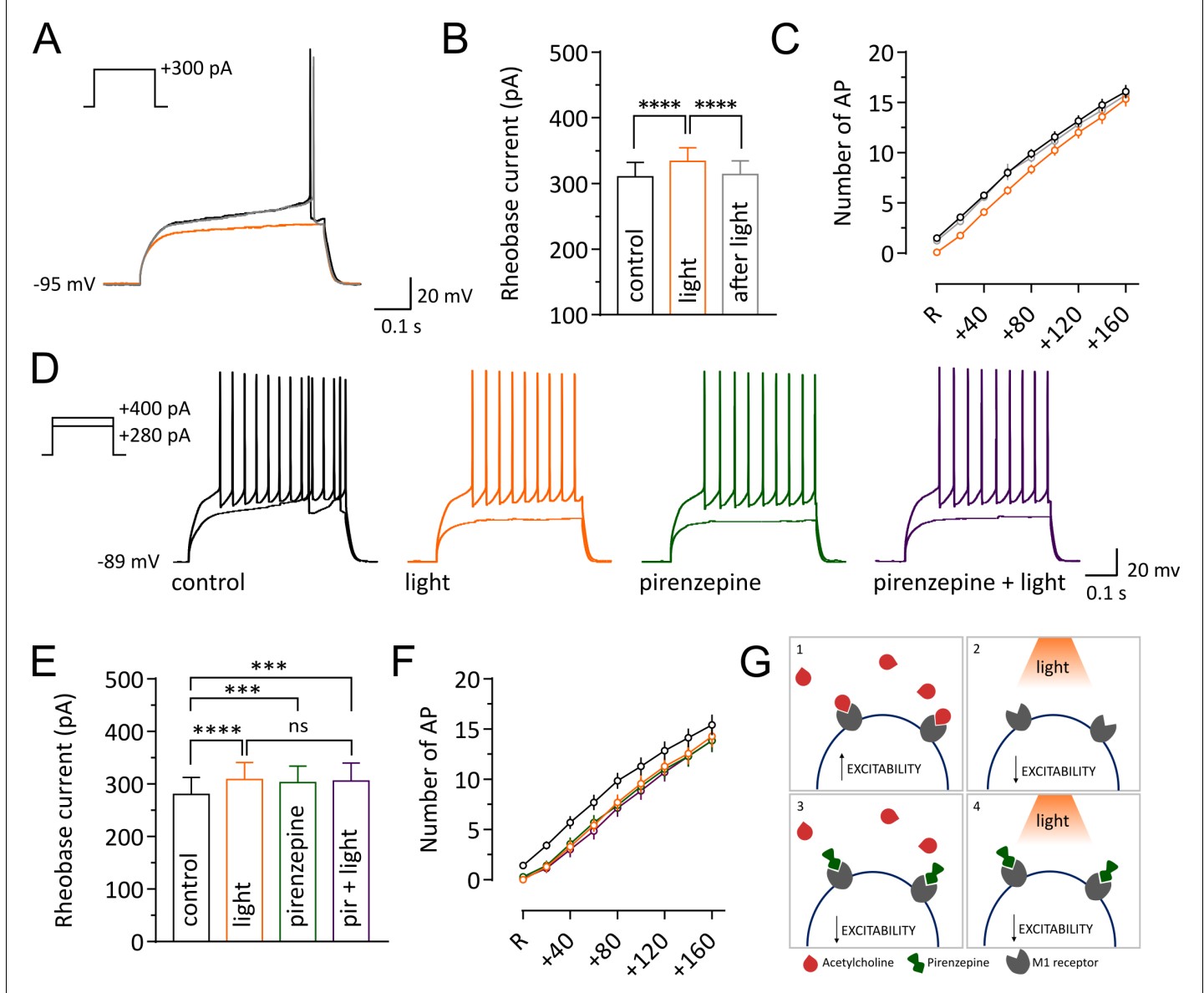

**Figure 4.** Light-induced pause in CINs decreases intrinsic excitability of SPNs via M1 receptor signaling. Representative traces of an SPN showing membrane responses to current injections before (black) during (orange) and after a pause (gray). (B, E) Pooled data showing average rheobase current. Before vs pause: p =<0.0001; pause vs after: p =<0.0001, N/n = 8/12 in (B), by Tukey's multiple comparisons test. Before vs pause: p =<0.0001; before vs pirenzepine: p=0.0005; before vs pirenzepine + light: p=0.0001; n = 7 in (E), by Tukey's multiple comparisons test. (C, F) Action potential numbers plotted against current injection (N/n = 8/12 in C; N/n = 6/7 in F). (D) Representative traces of an SPN showing membrane responses under different conditions. (G) Tonic level of acetylcholine control SPNs excitability via activation of muscarinic receptors (1). Light-induced pause lowers acetylcholine (ACh) levels reducing M1 activation and decreasing excitability of SPNs (2). M1 receptors selective antagonist pirenzepine decreases SPNs excitability (3). Pirenzepine occludes the effect of the pause (4). In (C) and (F) R represents the minimum current needed to trigger an action potential in basal conditions. Data are represented as mean ± SEM; ns = not significant. N = mice; n = cells.

DOI: https://doi.org/10.7554/eLife.32510.015

The following source data is available for figure 4:

**Source data 1.** This spreadsheet contains the intrinsic properties and firing responses for the individual neurons shown in *Figure 4*.
DOI: https://doi.org/10.7554/eLife.32510.016

SPNs, complementing previous studies that have focused on the effects of burst-pause or pause-burst sequences.

The subthreshold membrane potential fluctuations measured by in vivo whole-cell recording revealed that during a pause there was a hyperpolarizing shift of subthreshold membrane potential fluctuations, a slower transition between hyperpolarized and depolarized membrane potentials, and reduced duration of spontaneously occurring synchronous depolarizations. The underlying membrane potential fluctuations are seen during slow-wave sleep and anesthesia, but during wakefulness the membrane potential fluctuations are less stereotyped, with depolarizing synaptic events of variable amplitude occurring in temporally unstructured sequences (*Mahon et al., 2006*; *Sippy et al., 2015*). Understanding exactly how a pause would modulate these less stereotyped membrane potential fluctuations requires further whole cell recording experiments in awake animals.

Our finding that pauses in CIN firing cause inhibition of SPN firing complements an earlier study (*English et al., 2011*) that showed a decrease in firing of SPNs after an optogenetically induced pause-burst sequence. The inhibitory effect reported in this earlier study was synchronous with the rebound increase in firing that often occurs after an optogenetically induced pause. Taken together with our findings, there thus appear to be two successive inhibitory effects related to the pause-burst sequence. First, our data show a pause mediated inhibition caused by interruption of cholinergic excitation of SPNs. Second, *English et al. (2011)* have shown a burst-mediated inhibition caused by activation of GABAergic inputs to SPNs from other neurons excited by acetylcholine.

The onset of the inhibitory effect on SPNs of the pause in CINs was statistically significant 400 ms after the onset of the pause, both in the group where the pause lasted 500 ms and in the group where the pause duration was 1 s. Consistent with this delayed effect, *English et al. (2011)* reported no inhibition of SPNs during a pause of 200 ms duration. The delayed onset of inhibition that we observed at 400 ms could not have been detected in this 200 ms timeframe. In an additional group of 3 cells, *English et al. (2011)* also did not detect an inhibition of MSNs during a 1 s exposure to light. The detection of inhibitory responses in cells with low firing rates and a high coefficient of variation in firing times is challenging, and in a small sample it is not possible to rule out an inhibitory effect. The absence of pause-related inhibition in the cells exposed to 1 s light in the earlier study might also reflect differences in the power output of the optical fibre (4–10 mW versus 10–30 mW output) and the diameter – 200 μm versus 125 μm etched down to 50 μm used in *English et al. (2011)*.

The use of pauses of longer duration enabled the effect of the pause to be measured in isolation from the effects of any prior or subsequent rebound firing of CINs. In the natural firing patterns of CINS, the duration of the pauses varies considerably in different behavioural situations. Pauses occurring in response to sensory cues paired with motivationally significant events acquire a pause duration in the range of 200–300 ms (*Aosaki et al., 1995*; *Morris et al., 2004*). In anaesthetized animals pauses of duration 380–800 ms have been reported in response to a light stimulus (*Schulz et al., 2011*). In vivo, spontaneous firing activity is broken up by pauses that vary in duration up to one second (*Sharott et al., 2012*). Thus, pauses of similar duration to those used in the current experiments occur naturally as well as in controlled behavioural paradigms. The use of longer duration pauses made it possible to distinguish effects of pauses from rebound firing. In our juxtacellular recordings both 0.5 s and 1 s light pulses showed a significant reduction of SPNs firing in the group average starting at 400 ms. Single cell analysis showed that during a light pulse of 1 s, 64% of the SPNs displayed a significant inhibition in their firing rate. We did not detect, however, any significant effect on the firing of individual SPNs in response to 0.5 s light. These results thus leave an open question about the modulatory role of short pauses of CINs (<400 ms) on the activity of SPNs and the striatal processing that occurs in response to sensory cues paired with motivationally significant events. We speculate that short pauses may exert their effects mainly by rebound increases in firing, acting on nicotinic receptors on GABA interneurons as shown by *English et al. (2011)*, while longer pauses may engage a dual mechanism involving pause-related removal of muscarinic excitation via M1 receptors, in addition to GABA inhibition activated on the rebound.

We used NpHR to optogenetically silence CINs during 0.5 and 1 s pauses for recording responses of SPNs in juxtacellular configuration, and changes in membrane potential fluctuations in cells recorded in whole-cell mode during 5 s pauses. Light activation of NpHR caused effective silencing of CINs, as shown by measurements of action potential firing at the soma in our recordings. One caveat for the use of the inward Cl- pump - especially during longer pulses - is the increased

rebound firing, evident at the light offset, which may intensify the naturally occurring rebound firing that often follow a pause in CIN firing (*Mahn et al., 2016*; *Raimondo et al., 2012*). Although in the current experiments the main focus is on the effects during the period of the silencing rather than at the time of the rebound, it may be helpful to use other tools in parallel in future experiments to avoid this possibility.

Cell by cell analysis of the effects of the CINs pause on SPNs showed that while the group average showed a significant decrease in firing rate during the 1 s pause, a fraction of the neurons in the group did not show an individually significant decrease in firing rate. There are several possible explanations for why some SPNs did not respond to optogenetic manipulation of CINs. It is theoretically possible that not all SPN neurons in the sample express M1 receptors. However, this seems unlikely given that nearly all medium spiny neurons have M1 receptor messenger RNA (*Yan et al., 2001*) and express M1 receptor protein (*Hersch et al., 1994*). The remaining possibilities include that the non-responding SPNs were beyond the reach of the axonal arborization of a pausing CIN; or in a location where CINs did not express the neuronal silencer NpHR.

The present results add new knowledge – the inhibitory effects on SPNs of the pause in CIN firing – to existing knowledge concerning the effects of burst-pause or pause-burst sequences. We observed inhibition caused by the pause, occurring before the onset of the burst, after a delay of approximately 400 ms. We consider the most consistent interpretation to be that the clearance of acetylcholine levels from the synaptic cleft by acetylcholinesterase is the crucial factor for the onset of the inhibition we observed. It has been an open question whether the dynamics of acetylcholine clearance are fast enough for acetylcholine concentration to track momentary pauses in firing. If clearance were not extremely fast, the extracellular space would act as a capacitor and smooth out the effects of pauses. Clearance by hydrolysis of acetylcholine is catalyzed by acetylcholinesterase (AChE). The high levels of expression of AChE in the striatum (*Zhou et al., 2001*), and the extremely fast kinetics of this enzyme (*Quinn, 1987*), imply rapid clearance and sensitivity to dynamic fluctuations in CIN firing rate. However, current techniques for measuring acetylcholine concentration have insufficient temporal resolution to detect dips during pauses (*Mattinson et al., 2011*). The observed effects of a pause in CIN firing on SPNs, are thus the first direct evidence that pauses in CIN firing are meaningful signals to the postsynaptic SPNs.

Analysis of the firing of SPNs in the period immediately after the pause in CIN firing did not show an additional inhibition in the present study. Thus, the rebound firing of CINs that occurred in this period appeared to cause no additional inhibition relative to the firing rate reached during the exposure to light. The recovery to baseline levels after the 1 s pause, however, was relatively slow compared to the recovery after the 0.5 s pause. After the 1 s pause there was no significant increase in firing until 700 ms after the end of the light exposure, consistent with a second inhibitory process. During longer pauses (1 s), acetylcholine levels would be expected to have decreased below baseline levels, leading to the inhibition reported in the present study. Inhibitory effects of the rebound burst might be expected to add to the pause-related inhibition. However, these might not be apparent in the measurements due to a floor effect. That is, the firing rate of the SPNs is already low at the end of the longer pause, and further inhibition at this point would not produce significant effects. An implication of our findings is therefore that the effects of a burst following a pause may depend on the duration of the pause.

During natural firing patterns, CIN excitation preceding and following a pause can be expected to result in a complex interplay of the temporal dynamics of acetylcholine level changes with the dynamics of muscarinic and nicotinic receptors in SPNs, presynaptic glutamatergic inputs, or GABAergic interneurons. In vivo, there is often an excitation before the pause (*Matsumoto et al., 2001*). A burst preceding a pause was not studied in the current experiments, which started with the optogenetically-induced pause. Previously, *Witten et al. (2010)* used optogenetically induced excitation of CINs and found that a 10 Hz burst caused inhibition of SPNs. In that case the first burst in the burst-pause sequence would be expected to initiate a complex sequence of events. The burst would have excitatory effects on neuropeptide Y–expressing neurogliaform neurons mediated by nicotinic receptors, causing fast GABA inhibition (*English et al., 2011*; *Faust et al., 2015*; *Faust et al., 2016*) and excitatory presynaptic effects leading to release of GABA from dopamine terminals on a slower timescale (*Nelson et al., 2014*). During the initial burst, muscarinic receptors would also be activated causing presynaptic inhibition of cortical input by M2 (*Ding et al., 2010*)

and increased excitability by M1 activation (*Lv et al., 2017*) Activation of nicotinic receptors on dopamine terminals would also cause dopamine release (*Threlfell et al., 2012*).

Using a standard protocol to assess the short term synaptic changes typical of corticostriatal synapses, we observed that during a pause such protocol produced an increase in the first EPSP of the train, while the total facilitation measured as the ratio between the last to first pulse was decreased, thus defining a temporal window in which corticostriatal integration was attenuated. We did not investigate the mechanism underlying this effect. It is possibly related to muscarinic receptors that are expressed at pre-and postsynaptic sites on corticostriatal synapses on SPNs. A brief pause of tonic activation of presynaptic M2/M4 receptors controlling glutamate release (*Pakhotin and Bracci, 2007*; *Pancani et al., 2014*) might lead to the increased amplitude of the first EPSP. Another possibility is a postsynaptic effect of reduced M1 receptor activation (*Ding et al., 2010*; *Shen et al., 2007*). Although other mechanisms can be at play, the phenomenon observed suggests that the pause might activate a filter, enhancing the signal-to-noise ratio, effectively reducing the ability of spatial summed excitatory inputs to reach action potential threshold and activate the postsynaptic SPN. However, further experiments would be required to determine the physiological basis for the pause-related perturbation in synaptic integration.

Our recordings from acute slices show that the pause induced an increase in spike threshold and a decrease in excitability of SPNs. Both effects are occluded by pirenzepine, an inhibitor of M1 and M4 receptors. Thus, reduced muscarinic receptor activation due to decreased acetylcholine concentration during a pause in CIN firing may be one mechanism for the expression of this inhibition on SPNs activity. Acetylcholine activation of the M1 receptor excites SPNs by reducing membrane $K^+$ conductances (*Hsu et al., 1996*), modulating a transient $K^+$ conductance (*Akins et al., 1990*), and increasing NMDA receptor mediated currents and persistent $Na^+$ currents (*Calabresi et al., 1998*, *2000*). Conversely, a decrease in M1 receptor activation as occurs during a pause, would be expected to increase the membrane $K^+$ conductances, hyperpolarizing the membrane potential, and decrease NMDA currents, causing shorter duration of the UP state (*Plotkin et al., 2011*). Hence, the present findings are compatible with known actions of muscarinic M1 receptors on SPNs.

The inhibitory effect of the pause suggests a novel mechanism for the modulatory function of CINs in striatal function. The activity of CINs in awake animals is related to behavioral contexts such as reward probability (*Morris et al., 2004*; *Shimo and Hikosaka, 2001*), stimulus location (*Ravel et al., 2006*) or behavioral context and current state (*Lee et al., 2006*; *Stalnaker et al., 2016*). Ensembles of SPNs are recruited by cortical inputs causing transitions to firing activity. A pause in CIN firing would inhibit the SPNs within its axonal arbor, and favour a transition to a more hyperpolarized membrane potential, transiently decreasing the neuronal output of the ensemble of SPNs. In the context of inhibitory interactions among SPNs (*Tunstall et al., 2002*), this might favour cortical information transfer in a different ensemble of SPNs in the domain of tonically firing CINs. Acetylcholine released by a burst preceding a pause would have presynaptic effects on dopamine terminals (*Cragg, 2006*) and experiments using optogenetic stimulation show that additional CIN spikes cause phasic release of dopamine by activating presynaptic nicotinic receptors on dopamine terminals (*Threlfell et al., 2012*). These nicotinic receptors rapidly desensitize in increased acetylcholine and may be expected to regain their sensitivity to acetylcholine after a pause, facilitating further dopamine release. Recovery from desensitization during the pause depends on the duration of the exposure to acetylcholine (*Reitstetter et al., 1999*). Partial recovery begins on a subsecond timescale (*Yu et al., 2009*) consistent with the duration of a pause. During the pause, acetylcholine clearance occurs allowing recovery of nicotinic receptors to continue. At the same time, the stimulation is removed from muscarinic receptors leading to the inhibition that we describe. The net effect will depend on the acetylcholine levels reached during the burst, the rate of clearance, and the duration of the subsequent pause. The sequence would enable dopamine-dependent plasticity of the corticostriatal synapses active at the time of the pause. Such a mechanism may be important for understanding set shifting in behavioral flexibility, and loss of flexibility in neuropsychiatric diseases such as Parkinson's disease (*Ztaou et al., 2016*), dystonia (*Pisani et al., 2007*) and schizophrenia (*Lieberman et al., 2008*).

## Materials and methods

### Mice, Virus injections

Male heterozygous B6;129S6-*Chat*<sup>tm2(cre)Lowl</sup>/J mice (ChAT-cre; The Jackson Laboratory) were used in the present study. All experimental procedures were performed in accordance with and approved by the Okinawa Institute of Science and Technology Animal Care and Use Committee. Adeno-associated virus (AAV) encoding the neuronal silencer NpHR (AAV5-DIO-NpHR-eYFP) was obtained from the University of Pennsylvania (Gene Therapy Program, Pennsylvania, USA) or University of North Carolina (UNC vector core, USA). At postnatal day 14 (P14) we made a 300 nl unilateral stereotaxic guided injection in ChAT-cre mice. The injection volume and flow rate were controlled with an injection pump (World Precision Instruments, Sarasota, USA). Viral injections were targeted to the dorsal striatum (AP +0.7, ML ±1.5 relative to bregma, DV −1.7 relative to dura). Mice were anesthetized with vaporized isoflurane (Muromachi, Japan) and returned to the home cage with the doe after the end of the surgery. Post-surgery, mice were given a single dose of Carprofen (Rimadyl, 0.5 mg/kg IP) and Bupurenorphin (Lepetane, 0.05 mg/kg IP) for pain alleviation.

### Slice electrophysiology

Injected ChAT-cre mice aged 7–10 weeks old were anesthetized with isoflurane and decapitated. The brain was quickly removed and rested for 30 s in ice-cold oxygenated NMDG cutting solution containing (in mM): 92 NMDG, 2.5 KCl, 1.25 $NaH_2PO_4$, 30 $NaHCO_3$, 20 HEPES, 25 glucose, two thiourea, 5 Na-ascorbate, 3 Na-pyruvate, 0.5 $CaCl_2$·, 10 $MgCl_2$, and titrated to 7.2–7.4 pH with HCl. Slices (300 μm thick) containing the striatum were cut on a vibratome (VT1200S, Leica, Germany) on an oblique plane, at 45° between coronal and horizontal cuts to preserve corticostriatal connections. Slices were mounted on a porous membrane and incubated for 30 min at 34° C in oxygenated ACSF containing the following (in mM): 126.0 NaCl, 2.5 KCl, 2.0 CaCl2, 2.0 MgCl2, 18.0 NaHCO3, 1.25 NaHPO4, 10.0 glucose, then allowed to recover for at least 1 hr at room temperature before recording. Whole cell recordings were obtained from SPNs identified by size in the dorsal striatum in an area where fluorescent CINs expressing NpHR-eYPF could be visualized (OlympusBX51WI, DAGE-MTI IR-1000, and Olympus DP72). Pipets (4–6 MΩ) were pulled from P-97 (Sutter Instruments, Novato, CA) and filled with an intracellular solution containing the following (in mM): 119 K-MeSO4, 12 KCl, 1 MgCl2, 0.1 CaCl2, 10 HEPES, 1 EGTA, 0.4 Na-GTP, 2 Mg-ATP, (280–300 mOsm, pH 7.3 adjusted with KOH). Recordings were performed in a chamber perfused with ACSF at a rate of 3 ml/min and maintained at 32°C. Acquisition was done using Clampex 10.4, MultiClamp 700B amplifier and Digidata 1440A (Molecular Devices, San Jose, CA). For optogenetic manipulations, CINs expressing NpHR-eYFP were activated with short pulses of green light generated by a 530 nm LED (CAIRN, UK) under the control of the digital output of the amplifier. Current steps of 500 ms at increments of 20 pA were used to assess rhoebase current as well as the relationship between AP number and increasing current injection. Pirenzepine was bath applied for at least 5 min before recording the membrane responses to current step injections.

### In vivo electrophysiology

Mice between 5 and 8 weeks old were anesthetized with a ketamine/xylazine mix (100 and 10 mg/kg i.p., respectively, supplemented as needed injection during the course of the experiment) and fixed on a Kopf stereotaxic apparatus. The animals were kept warm (~37°C) for the whole duration of the surgery via a heating pad connected to a DC temperature controller provided with a feedback system (FHC Inc., Bowdoin, ME). An eye lubricant was applied to prevent corneal drying during the surgery. A stereomicroscope (LEICA M651) was used to locate the atlas coordinates, using bregma as a reference (*Paxinos and Franklin, 2001*). Two craniotomies were drilled between 0 and 1.5 mm anterior from bregma, to allow the insertion of the optic fiber and recording electrode, and between 1.8 and 2.2 mm for insertion of a bipolar stimulating electrode for experiments of corticostriatal plasticity. Electrophysiological recordings were obtained from dorsal striatum between 1,9 and 2.5 mm from dura using a Multiclamp 700B amplifier connected to a Digidata 1440A. Data were acquired with pClamp 10 (Molecular Devices), digitized at 20 kHz, filtered at 10 kHz, and analyzed offline with Clampfit 10.4 (Molecular Devices) or MATLAB as indicated. Activation of NpHR was achieved with a 100 mW, 593 nm laser (Shanghai Dream Lasers Technology Co., Ltd., China) via an optic fiber (200

μm, 0.22 NA) inserted into the dorsal striatum (+1 mm AP, 0 mm ML, −2.1 mm DV from dura) with a 26° angle. Light pulses were externally triggered using pClamp (Molecular Devices). The intensity of light stimulation was between 4–10 mW, measured at the tip of the fiber, outside of the brain. To measure the intensities of light we used a an optical power meter (model 1936 C, Newport, Irvine, CA, USA) equipped with a photodiode sensor (model 918D-UV-OD3, Newport, Irivine,CA, USA). For the activation of NpHR we used a square pulse with the following durations: 0.5 s, 1 s, 5 s. Whole-cell patch-clamp recordings were achieved using a standard blind-patch approach as previously described (*Margrie et al., 2002*). Electrodes used for whole-cell recordings had a resistance between 4.5 and 6 MΩ and were filled with a solution containing (in mM) 135 K-gluconate, 4 KCl, 10 HEPES, 10 sodium phosphocreatine, 4 Mg-ATP and 0.3 Na-GTP (290–300 mOsm, pH 7.2–7.3 adjusted with KOH). The intracellular solution was supplemented with 0.2% (wt/vol) biocytin for *post hoc* cellular identification and morphological reconstruction. Only SPNs that showed a typical bimodal distribution of the membrane potential and initial membrane potentials values $\leq -60$ mV in vivo were included in the study. At the end of each recording, injection of biocytin was achieved by applying positive current pulses (0.5–1 nA, 500 ms, for 5–10 min) through the bridge circuitry of the amplifier.

Juxtacellular recordings of striatal neurons were obtained using glass electrodes (5–7 MΩ) filled with Ringer's solution containing (in mM): 135 NaCl, 5.4 KCl, 5 HEPES, 1.8 CaCl2, 1 MgCl2, or intra-cellular solution. Juxtacellular units were high pass filtered at 300 Hz. Light durations of 0.5 and 1 sec s, were delivered at 50 s interval and repeated 10 times.

Spontaneous firing properties of striatal neurons in juxtacellular configuration were estimated during a 3–5 min period before any stimulation (*Figure 1—figure supplement 2*). Optically tagged single unit showing patterned pause-burst responses were classified as CINs. Cluster separation between CINs and SPNs was done using classically defined electrophysiological characteristics (*Isomura et al., 2013*; *Sharott et al., 2012*): spike waveform (half-width), firing frequency and regularity (see *Figure 1—figure supplement 2*). Firing frequency, interspike intervals and half-width were calculated using a detection threshold function in Clampfit. The units obtained in awake head fixed configuration showed similar clustering (*Figure 1—figure supplement 2C and D*) and were pooled together.

For plasticity experiments, a bipolar electrode was used to stimulate cortical inputs from the motor cortex area which send dense projections to the dorsal striatum (*Wall et al., 2013*). EPSPs were electrically evoked with trains of 10 stimuli delivered at 25 Hz (repeated every 30 s). To test the effect of a pause each train started 500 ms after the onset of the light pulse (total duration: 1 s). The EPSP amplitude was quantified by measuring the peak of the synaptic response obtained from the average trace of all the sweeps recorded. Values were normalized for the first EPSP amplitude.

UP and DOWN states were algorithmically detected using the strategy proposed for characterizing membrane potential fluctuations in electrophysiological data (*Reig and Silberberg, 2014*; *Seamari et al., 2007*). UP states that were at least 0.2 s in duration were considered. Data were further analyzed with MATLAB 2014.

Spontaneously occurring UP and DOWN states were analyzed before, during and after a pause in CINs firing, by sampling a window of 5 s in each condition. The duration of the light was 5 s, delivered at 30 s interval and repeated at least five times. UP and DOWN states occurring during a pause were compared to those occurring in the preceding 5 s and 10 s after the end of the light pulse, when CINs tonic firing recovered to baseline rate, based on observations from juxtacellular recordings. The average of all UP states occurring before (n = 334), during (n = 319) and after a pause (n = 325) revealed a slower slope of the UP states during light. To quantify the slope of the transitions from DOWN to UP state, UP states were center aligned and scaled to have equal baseline values. The slopes of the transition from DOWN to UP state was calculated from the best-fit line of a linear regression on the membrane potential values in the region between 10% to 90% of the peak of the UP state. The average peak potential was calculated by taking the middle 50% of the membrane potential values for each single UP state event.

Head-fixed awake in vivo electrophysiology experiments required a second surgery for implanting the head-bar after at least 6 weeks from virus injection. Mice were anaesthetized with gaseous iso-fluorane (Muromachi, Japan) and placed in a stereotaxic frame. The skull was cleaned to locate and mark bregma with an oil based marker and a head-bar (Phenosys, Germany) was glued to the skull with Super-bond (Sun Medical, Japan). Post-surgery, mice were given a single dose of Carprofen

(Rimadyl, 0.5 mg/kg IP) and Buprenorphin (Lepetane, 0.05 mg/kg IP) for pain alleviation. 3 days after implantation, mice were handled for at least three sessions as they habituated to head-fixation and jet-ball (PhenoSys, Germany). Mice were able to freely moving over the spherical ball and they all exhibited grooming, resting, or running behaviors during each session. After habituation, on the day of electrophysiology recording, a craniotomy was performed under ketamine/xylazine mix. Mice recovered from the anaesthesia before initiation of Juxtacellular recordings and optical stimulation of CINs expressing NpHR-eYFP were obtained using same procedure as reported for anesthetized experiments.

## Immunohistochemistry and image analysis

To recover the morphology of neurons filled with biocytin during in vivo whole-cell patch clamp recordings, brains were removed after completion of electrophysiology experiments and fixed for at least one week at 4°C in a solution containing 4% paraformaldehyde (PFA) in 1X phosphate buffer saline (PBS), pH 7.0. Coronal slices (60 µm thick) were prepared with a vibratome (VT1000S, Leica), washed with 0.3% Triton-X-100 in 1X PBS and incubated overnight at 4°C in Alexa Fluor 594 streptavidin conjugate (diluted 1:1000; Molecular Probes, Japan, S11227) in the same solution. To analyse the extent of the viral spread in cholinergic interneurons, slices were incubated overnight with the primary antibody goat anti-Chat (diluted 1:1000; Millipore, AB144P). Slices were then washed in 1X PBS and incubated overnight at 4°C in 1:1000 Alexa Fluor 594 (diluted 1:1000; Invitrogen, Japan, A11058) in the same solution. Immunofluorescence data of mounted coverslips were acquired with a confocal laser scanning microscope (Zeiss LSM 780) using Argon 488 nm and DPSS 561 nm lasers and 10x/0.45 and 20x/0.8 Plan-Apochromat lens controlled by Zen 2012 software.

Quantification of colocalization of ChAT and NpHR-eYFP was performed using four alternate sections per mouse (N = 4), in a range between +1.2 to+0.8 mm (AP) from Bregma, corresponding to the coordinates used for the placement of the fiber optic. Given the lack of a clear boundary between dorsal and ventral striatum we objectively defined the dorsal region of the striatum by drawing a horizontal line at ~50% of the area of the striatum. Within this region we identified and counted a total of 903 neurons: ChAT$^+$, NpHR-eYFP$^-$=387; ChAT$^-$, NpHR-eYFP$^+$=5; and ChAT$^+$, NpHR-eYFP$^+$=511.

## Drugs

The following drugs were used: pirenzepine dihydrochloride (1071, Tocris). Aliquots of stock solutions were prepared and frozen at −20°C. All other chemicals were from either Sigma-Aldrich, Wako, or Nacalai Tesque, Japan.

## Statistics

Statistical analysis was performed with GraphPad Prism seven software. Values are given as mean ± SEM of n experiments. D'Agostino and Pearson normality test, Shapiro-Wilk normality test and KS normality test were used to determine if the values in our data came from a Gaussian distribution. All data sets passed at least two out of the three normality test, and thus we used parametric tests. Repeated measure two-way ANOVA for analysis of light-drug interaction in acute slices experiments were matched by sub-column and spread across a row with Sidak's multiple comparisons test. Repeated measure with one-way ANOVA with Tukey's multiple comparison test were used to compare before, during and after pause in *Figure 2D*, *Figure 2E*, *Figure 2H* as well as *Figure 4B*. For *Figure 4E*, we also used a repeated measure with one-way ANOVA followed by Tukey's multiple comparison test to compare rheobase changes between treatments (control, light, pirenzepine, pirenzepine with light). We used an Ordinary One-way ANOVA for *Figure 2F*. Student's paired *t*-test were used for all other comparisons between two groups. No statistical tests were run to predetermine the sample size, and blinding and randomization were not performed. Probability threshold for statistical significance in these analyses was p<0.05.

To analyse the effects of light exposure on the firing rate of SPNs we used a minimal criterion of two S.D. difference from the mean. To take in account the low firing rate and high coefficient of variation in the spike timings, which presents a challenge for analyses to test for statistical significance, we used a standard method of analysis based on the cumulative sum of residuals (CUSUM), to identify the point at which deviations from baseline became significant (*Ellaway, 1978*; *Page, 1957*). To

calculate the CUSUM, spikes were binned into a PSTH. The mean firing rate (μ) within a reference window of $n_1$ bins immediately before the onset of the light, and the S.D. of the firing rate ($\sigma$) were calculated. The CUSUM was calculated by subtracting μ from each bin. The cumulated successive differences were calculated using the formula for the $r^{th}$ CUSUM (*Davey et al., 1986*; *Ellaway, 1978*) of:

$$S_r = \sum_{i=1}^{r}(x_i - \mu)$$

A change in firing rate was considered significant if the observed deviation of the CUSUM from the mean had a probability predicted from a Poisson distribution of p<0.01 (for group data) or p<0.05 (for single neurons). The critical values for the $i^{th}$ bin were calculated using the derivation of *Imamura and Onoda (1983)*:

$$P_r\{S_i \geq 2.33\,\sigma\sqrt{i-n_1}\} = 0.01, \text{ or}$$

$$P_r\{S_i \geq 1.65\,\sigma\sqrt{i-n_1}\} = 0.05$$

The CUSUM was calculated using a reference window of 1 s immediately prior to the light on period. To analyse the recovery after the light was turned off, in the group exposed to a 1 s illumination, the CUSUM was calculated using a reference window of 0.5 s immediately prior to the light off. Bins with frequencies more than 2 s.D.s below the mean firing rate of the baseline were considered significantly different from baseline if they occurred after the CUSUM crossed the significance level.

## Acknowledgements

This study was supported by the Okinawa Institute of Science and Technology Graduate University and the Human Frontier Science Program RGP0048/2012 – NIV. We thank Andrew Liu for help with genotyping.

## Additional information

### Funding

| Funder | Grant reference number | Author |
| --- | --- | --- |
| Okinawa Institute of Science and Technology Graduate University | | Jeffery Wickens |
| Human Frontier Science Program | HFSP - RGP0048/2012 - NIV | Jeffery Wickens |

The funders had no role in study design, data collection and interpretation, or the decision to submit the work for publication.

### Author contributions

Stefano Zucca, Conceptualization, Formal analysis, Investigation, Methodology, Writing—original draft, Writing—review and editing; Aya Zucca, Conceptualization, Formal analysis, Investigation, Methodology, Writing—original draft, Project administration, Writing—review and editing; Takashi Nakano, Software, Formal analysis, Methodology; Sho Aoki, Resources, Methodology, Writing—review and editing; Jeffery Wickens, Conceptualization, Supervision, Project administration, Writing—review and editing

### Author ORCIDs

Stefano Zucca (iD) http://orcid.org/0000-0002-8781-3295
Aya Zucca (iD) http://orcid.org/0000-0001-8613-8315
Jeffery Wickens (iD) http://orcid.org/0000-0002-8795-1209

## Ethics

Animal experimentation: This study was performed in strict accordance with the recommendations in the Guidelines for Proper Conduct of Animal Experiments of the Science Council of Japan. All experimental procedures were performed in accordance with a protocol approved by the Animal Care and Use Committee (ACUC) of the Okinawa Institute of Science and Technology Graduate University (Protocol #2014-089). All surgery was performed under general anesthesia with vaporized isoflurane and post-surgery pain alleviation. Every effort was made to minimize suffering.

## Decision letter and Author response

Decision letter https://doi.org/10.7554/eLife.32510.019
Author response https://doi.org/10.7554/eLife.32510.020

# Additional files

## Supplementary files

• Transparent reporting form
DOI: https://doi.org/10.7554/eLife.32510.017

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
