## [Decision Letter]

Thank you for submitting your article "Pauses in cholinergic interneuron firing exert an inhibitory control on striatal output in vivo" for consideration by *eLife*. Your article has been reviewed by three peer reviewers, and the evaluation has been overseen by Michael Frank as the Senior Editor and Naoshige Uchida as the Reviewing Editor. The following individual involved in review of your submission has agreed to reveal her identity: Veronica Alvarez (Reviewer #1).

The reviewers have discussed the reviews with one another and the Reviewing Editor has drafted this decision to help you prepare a revised submission.

Summary:

In this brief report, the authors tested the impact of silencing striatal cholinergic interneurons (CINs) on the firing of striatal spiny projection neurons (SPNs) in vivo and in vitro. They selectively expressed optogenetic tools (NpHR) in the cholinergic interneurons, and used yellow light stimulation to silence the firing of striatal cholinergic interneurons. During the light stimulation, they observed that striatal SPNs have reduced excitability: increased time in down-state, reduced time in up-state, reduced action potentials, and increased time transitioning from down-state to up-state. And consistent with previous observations, it is likely caused by reduced M1 receptor activation, and subsequently reduced intrinsic excitability caused by M1 receptor modulation. The reviewers appreciated that the in vivo whole cell recordings are technically challenging, and the data presented in the paper are generally of high quality.

It has been widely observed that CINs display a pause of firing in response to salient stimuli. However, its significance and how it affects the activity of other neurons remain to be determined. Although this issue has been addressed, at least in part, in previous studies, the result of the present study differs from previous conclusions (e.g. Witten et al., 2010 reported an increase in SPN firing during CIN inhibitions and a decrease during CIN excitations). The data presented in the present study provide deeper insights into how CIN pauses affect the activity of SPNs.

However, the reviewers raised various concerns regarding the data analysis and the validity of some conclusions. We, therefore, would like to see your response.

Essential points:

1) In animals that are not engaged in overt behaviors, SPNs fire at very low rates (less than 1 spike per second) and in a highly irregular (sporadic) manner. Such firing patterns present a challenge for analyses designed to detect significant decreases in firing. The authors have used a highly unusual metric to define a decrease in firing, namely dipping under 35% of the mean baseline firing. Equally worrying, there is no account of how many of the SPNs tested actually exhibited a significant decrease in firing. The current analyses are not rigorous or compelling. There is insufficient evidence to support the claim of a "powerful inhibition", but this issue goes beyond semantics. The data should be re-analyzed (on the basis of single neurons, as well as groups), using standard techniques for normalization and a more stringent threshold for significance, e.g. +/- 2 SDs of mean baseline firing. The proportion of tested neurons that reach criteria should be reported.

2) The authors delivered light pulses with durations of 0.5, 1 or 5 seconds, and subsequently inhibited (silenced) some CINs for similar durations. One reviewer raised a concern that these durations are artificial because the evidence in the literature indicates that the typical pause duration of TANs (putative CINs) is 200-250 milliseconds although a different reviewer pointed out that some literatures show CIN pauses that last more than 500 ms. However, the reviewer raised the issue that CIN pauses are often preceded or followed by burst firing, and these CIN excitations surrounding a pause may obscure the main observations of CIN pauses inhibiting SPN activity.

To remedy these issues, the reviewers suggest the following. First, the authors should more carefully discuss the relevant literatures and justify the durations of optogenetic inhibitions. It is preferable if SPN responses to light pulses of shorter durations (200-250 ms) were tested but this issue could be remedied if the authors can justify longer durations of inhibition based on the literature. Second, the reviewers noticed that optogenetic inhibition of CINs was often followed by rebound excitation of CINs. This appears to give the opportunity to address, at least in part, the issue of surrounding excitations. Specifically, please analyze the firing of SPNs immediately following CIN inhibitions. Based on the result, the authors may be able to discuss the impact of CIN's rebound excitations. The authors noted that excitation preceding a CIN pause cannot be addressed by addressing this, but this issue can be beyond the scope of the present study. Nonetheless, the reviewers would like to see a careful discussion on this issue (the potential effects of CIN excitations both preceding and following a pause).

Related to the issue of CIN rebound excitations, one reviewer raised the following issue: After NpHR silencing, CINs show prominent rebound firing with sustained enhanced firing that lasts for hundreds of milliseconds. If the proposed mechanism were true, one would expect that muscarinic tone would be significantly enhanced, and SPN firing would be increased. However, the data did not show such effect. What could be the explanation?

3) The wider relevance of studying the effects of 5 second pulses of light on the UP and DOWN states of SPNs in anesthetized mice is unclear. This is because SPNs in awake behaving animals do not exhibit such stereotyped fluctuations in membrane potential (see, for example, Mahon et al., PMID:17135420 and Sippy et al., PMID:26439527). A key question thus remains answered; what is the effect of CIN pauses on SPN membrane potentials and firing when animals are not anesthetized? The reviewers suggest that the authors reduce the tone of discussing UP and DOWN states.

4) Figure 3 shows that integration of cortical inputs was severely impaired when CINs were silenced by NpHR. However, such perturbation in synaptic integration is unlikely caused by reduction in M1 modulation. The authors should perform similar experiments in brain slice with M1 receptor antagonist to further validate the finding, or reduce the tone of the involvement of M1 receptors.

5) The authors should discuss the differences from previous studies more explicitly in order to help identify the novelty of the present work.

---

## [Author Response]

Essential points:1) In animals that are not engaged in overt behaviors, SPNs fire at very low rates (less than 1 spike per second) and in a highly irregular (sporadic) manner. Such firing patterns present a challenge for analyses designed to detect significant decreases in firing. The authors have used a highly unusual metric to define a decrease in firing, namely dipping under 35% of the mean baseline firing. Equally worrying, there is no account of how many of the SPNs tested actually exhibited a significant decrease in firing. The current analyses are not rigorous or compelling. There is insufficient evidence to support the claim of a "powerful inhibition", but this issue goes beyond semantics. The data should be re-analyzed (on the basis of single neurons, as well as groups), using standard techniques for normalization and a more stringent threshold for significance, e.g. +/- 2 SDs of mean baseline firing. The proportion of tested neurons that reach criteria should be reported.

We have reanalyzed the data using standard techniques and a stringent threshold for significance. The lines on the peristimulus time histogram now show 2 S.D.s from the mean. To statistically test at what point changes became significant, we calculated the cumulative sum of residuals of the observed deviation of the firing rate from the mean, and tested if the difference exceeded a critical value predicted from a Poisson distribution. We have detailed this approach in the Materials and methods section as follows:

“To analyse the effects of light exposure on the firing rate of SPNs we used a minimal criterion of 2 S.D. difference from the mean. […] Bins with frequencies more than 2 S.D.s below the mean firing rate of the baseline were considered significantly different from baseline if they occurred after the CUSUM crossed the significance level.”

We also analyzed the proportion of neurons that showed significant changes individually. For this analysis we applied the 2 S.D. criterion and the CUSUM as detailed in the Materials and methods section (above). We report the findings in the Results section as follows:

“The light-induced pause in CIN firing caused inhibition of firing activity in SPNs (Figure 1 and Figure 1) which dropped below 2 S.D. from the mean firing frequency (Figure 1). […] These results indicate that a pause in CIN firing of 1 s duration significantly reduces the firing of SPNs in vivo.”

We have deleted adjectives such as “powerful” from the text and instead report statistical significance levels.

In the case of the 0.5 s light exposure, this data is also reanalyzed and is now presented in the supplementary materials because we felt it did provide additional information.

2) The authors delivered light pulses with durations of 0.5, 1 or 5 seconds, and subsequently inhibited (silenced) some CINs for similar durations. One reviewer raised a concern that these durations are artificial because the evidence in the literature indicates that the typical pause duration of TANs (putative CINs) is 200-250 milliseconds although a different reviewer pointed out that some literatures show CIN pauses that last more than 500 ms. However, the reviewer raised the issue that CIN pauses are often preceded or followed by burst firing, and these CIN excitations surrounding a pause may obscure the main observations of CIN pauses inhibiting SPN activity.To remedy these issues, the reviewers suggest the following. First, the authors should more carefully discuss the relevant literatures and justify the durations of optogenetic inhibitions. It is preferable if SPN responses to light pulses of shorter durations (200-250 ms) were tested but this issue could be remedied if the authors can justify longer durations of inhibition based on the literature.

We have carefully discussed the relevant literature concerning the duration of pauses. We considered pauses occurring during spontaneous firing activity as well as stimulus-evoked pauses. We added new discussion of relevant papers showing that longer duration pauses occur in both kinds of activity, see Discussion as follows:

“The use of pauses of longer duration enabled the effect of the pause to be measured in isolation from the effects of any prior or subsequent rebound firing of CINs. […] In any case, it is important to distinguish effects of pauses from rebound firing, and the available methods for detecting the effects of the pause require longer durations.”

Second, the reviewers noticed that optogenetic inhibition of CINs was often followed by rebound excitation of CINs. This appears to give the opportunity to address, at least in part, the issue of surrounding excitations. Specifically, please analyze the firing of SPNs immediately following CIN inhibitions. Based on the result, the authors may be able to discuss the impact of CIN's rebound excitations. The authors noted that excitation preceding a CIN pause cannot be addressed by addressing this, but this issue can be beyond the scope of the present study. Nonetheless, the reviewers would like to see a careful discussion on this issue (the potential effects of CIN excitations both preceding and following a pause).Related to the issue of CIN rebound excitations, one reviewer raised the following issue: After NpHR silencing, CINs show prominent rebound firing with sustained enhanced firing that lasts for hundreds of milliseconds. If the proposed mechanism were true, one would expect that muscarinic tone would be significantly enhanced, and SPN firing would be increased. However, the data did not show such effect. What could be the explanation?

We have analyzed the firing of SPNs immediately following CIN inhibition when there is a brief period of increase in CIN firing (rebound). During and after the rebound period the firing of SPNs stayed 2 S.D. below the mean and did not show further significant inhibition. By 700 ms after the light off the firing rate of SPNs had increased significantly above the last 500 ms of the light on period, however it was still more than 2 S.D. below the mean at this point. Thus, the increase in acetylcholine associated with the rebound firing that follow the pause seems to be associated with a second inhibitory effect.

Regarding the explanation for why SPN firing is not increased after the light is turned off, we suggest that this may be due to a second inhibitory effect and a paragraph referencing the relevant literature has been added to the Discussion as follows:

“Analysis of the firing of SPNs in the period immediately after the pause in CIN firing did not show an additional inhibition in the present study. […] That is, the firing rate of the SPNs is already low at the end of the longer pause, and further inhibition at this point would not produce significant effects. An implication of our findings is therefore that the effects of a burst following a pause may depend on the duration of the pause.”

3) The wider relevance of studying the effects of 5 second pulses of light on the UP and DOWN states of SPNs in anesthetized mice is unclear. This is because SPNs in awake behaving animals do not exhibit such stereotyped fluctuations in membrane potential (see, for example, Mahon et al., PMID:17135420 and Sippy et al., PMID:26439527). A key question thus remains answered; what is the effect of CIN pauses on SPN membrane potentials and firing when animals are not anesthetized? The reviewers suggest that the authors reduce the tone of discussing UP and DOWN states.

We have reduced the tone of discussion of UP and DOWN states and explained that such stereotyped fluctuations in membrane potential do not occur in awake animals in the Discussion as follows:

“The subthreshold membrane potential fluctuations measured by in vivo whole-cell recording revealed that during a pause there was a hyperpolarizing shift of subthreshold membrane potential fluctuations, a slower transition between hyperpolarized and depolarized membrane potentials, and reduced duration of spontaneously occurring synchronous depolarizations. […] Understanding exactly how a pause would modulate these less stereotyped membrane potential fluctuations requires further whole cell recording experiments in awake animals.

4) Figure 3 shows that integration of cortical inputs was severely impaired when CINs were silenced by NpHR. However, such perturbation in synaptic integration is unlikely caused by reduction in M1 modulation. The authors should perform similar experiments in brain slice with M1 receptor antagonist to further validate the finding, or reduce the tone of the involvement of M1 receptors.

We agree that further experiments are needed. We have reduced the tone of the discussion concerning the involvement of M1 receptors in such mechanism throughout the text.

5) The authors should discuss the differences from previous studies more explicitly in order to help identify the novelty of the present work.

We have added the following paragraph in the Discussion to address this point:

“Based on in vivo whole cell recordings we show that optogenetically-induced pauses in CIN firing have direct and reproducible inhibitory effects on SPN activity. Pauses in CIN firing caused hyperpolarization of subthreshold membrane potential fluctuations measured by in vivo whole-cell recording and caused a significant decrease in the firing rate of SPNs measured using in vivo juxtacellular recordings. […] We believe these findings constitute the first direct evidence that the pause in CIN firing carries a meaningful signal to postsynaptic SPNs, complementing previous studies that have focused on the effects of burst-pause or pause-burst sequences.”